# A NEW PHOTORECEPTOR-INSPIRED CNN LAYER ENABLES DEEP LEARNING MODELS OF RETINA TO GENERALIZE ACROSS LIGHTING CONDITIONS

## ABSTRACT

As we move our eyes, and as lighting changes in our environment, the light intensity reaching our retinas changes dramatically and on multiple timescales. Despite these changing conditions, our retinas effortlessly extract visual information that allows downstream brain areas to make sense of the visual world. Such processing capabilities are desirable in many settings, including computer vision systems that operate in dynamic lighting environments like in self-driving cars, and in algorithms that translate visual inputs into neural signals for use in vision-restoring prosthetics. To mimic retinal processing, we first require models that can predict retinal ganglion cell (RGC) responses reliably. While existing state-of-the-art deep learning models can accurately predict RGC responses to visual scenes under steady-state lighting conditions, these models fail under dynamic lighting conditions. This is because changes in lighting markedly alter RGC responses: adaptation mechanisms dynamically tune RGC receptive fields on multiple timescales. Because current deep learning models of the retina have no in-built notion of light level or these adaptive mechanisms, they are unable to accurately predict RGC responses under lighting conditions that they were not trained on. We present here a new deep learning model of the retina that can predict RGC responses to visual scenes at different light levels without requiring training at each light level. Our model combines a fully trainable biophysical front end capturing the fast and slow adaptation mechanisms in the photoreceptors with convolutional neural networks (CNNs) capturing downstream retinal processing. We tested our model's generalization performance across light levels using monkey and rat retinal data. Whereas conventional CNN models without the photoreceptor layer failed to predict RGC responses when the lighting conditions changed, our model with the photoreceptor layer as a front end fared much better in this challenge. Overall, our work demonstrates a new hybrid approach that equips deep learning models with biological vision mechanisms enabling them to adapt to dynamic environments.

## 1 INTRODUCTION

A key problem in visual neuroscience is to generate models that can accurately predict how neurons will respond to visual stimuli. Along with their role in basic neuroscience, these models have applications in prosthetic devices and can form the basis for bio-inspired computer vision systems that aim to mimic the impressively robust functions of the human visual system. Use of machine learning models for such applications in neuroscience has become increasingly ubiquitous given their strong performance in computer vision applications like object recognition (Chollet, 2017; Simonyan & Zisserman, 2015; Krizhevsky et al., 2017). For example using convolutional neural networks (CNNs) to predict responses of neurons in visual cortex (Kindel et al., 2019; Cadena et al., 2017) and retina (McIntosh et al., 2016; Tanaka et al., 2019; Yan et al., 2022; Goldin et al., 2022) to visual stimuli. We focus here on the retina. Under carefully controlled experimental conditions, and with constant lighting conditions, CNN models can predict responses of retinal ganglion cells (RGCs, the "output" cells of the retina, whose axons form the optic nerve) to visual stimuli with high accuracy (McIntosh et al., 2016). In natural vision, however, lighting conditions are highly dynamic: the amount of light falling on the retina can change by several orders of magnitude at mul-

tiple timescales. For example, light input can change locally on a region of the retina by an order of magnitude in less than a second following rapid eye movements such as saccades; global light levels may fluctuate by an order of magnitude every few seconds on a cloudy day, as clouds pass between the observer and the sun. These changes in light level substantially alter RGC responses to visual stimuli (Tikidji-Hamburyan et al., 2015; Ruda et al., 2020; Idrees et al., 2020; 2022; Farrow et al., 2013), and pose the as-of-yet-unanswered question of whether CNNs can accurately predict RGC responses under these more challenging conditions.

To answer this question, we first tested the ability of conventional CNN models (i.e., the Deep Retina model of (McIntosh et al., 2016)) to predict RGC responses in the presence of changing light levels. These CNNs could accurately predict RGC responses when tested under the same lighting conditions under which they were trained but were unable to accurately predict responses under different lighting conditions.

To improve the performance of machine learning (ML) models in predicting RGC responses under varied lighting conditions, we created a new type of convolutional neural network layer as a front-end for CNNs. This fully trainable input layer mimics the transformation of light into electrical signals at the retina's photoreceptors, including the adaptation mechanisms that modulate photoreceptor response sensitivity and kinetics and by doing so enable the retina to operate under a wide and dynamic input range. This in-built adaptation allows the model to adapt to changing light levels locally, at each pixel location in the input, the same way our retinas do. CNNs with this new input layer could generalize to test lighting conditions very different from the lighting conditions under which they were trained.

We anticipate that our new photoreceptor-inspired CNN layer could have several significant impacts in various fields of vision science. First, our model can be used directly in vision-restoring prosthetics by enabling the translator algorithms to function under diverse lighting conditions, and thereby improve the operating range of these prosthetics. Second, our model can be used by visual neuroscientists to investigate dynamic retinal computations. Using the model as a front-end for algorithms that predict the responses of neurons in visual cortex, will enable those cortical models to operate under more naturalistic and more varied lighting conditions. Finally, because their local (pixel-by-pixel) adaptation to changing light levels can filter out image changes due to changing lighting and shadow, our photoreceptor-inspired CNNs could have substantial application for computer vision systems operating in outdoor environments. There, lighting changes and shadows are known to confuse existing object detection and recognition algorithms (Yadron & Tynan, 2016; Levin, 2018; Janai et al., 2020; Gomez-Ojeda et al., 2015; Kolaman et al., 2019) as they do not naturally separate image changes due to local lighting variations from image changes due to changing object content. Here, photoreceptor-inspired CNN models can discount disruptive events in a video stream, such as sudden large changes in the amount of light reflected off a tracked object due to shadows or changes in intensity of the light source.

## 2 RELATED WORK

### 2.1 DEEP LEARNING MODELS OF THE VISUAL SYSTEM

Deep learning approaches have become increasingly popular in modeling the visual system as they can describe neural responses to a visual scene markedly more accurately than linear-nonlinear (LN) models. The current state-of-the-art retina predictor, called Deep Retina (McIntosh et al., 2016), is based on a 2-layer CNN. This model takes as input a movie from which spatio-temporal features are extracted by the CNN layers, and outputs the spike rate of retinal ganglion cells. This model, when trained to predict tiger salamander retinal ganglion cell responses to spatiotemporal white noise stimuli, outperformed other models such as LN and the generalized linear models (GLMs). Such CNN based models of the retina have been used for describing responses to natural stimuli (McIntosh et al., 2016; Tanaka et al., 2019) and for explaining the underlying neural computations that lead to RGC responses (Tanaka et al., 2019; Yan et al., 2022; Goldin et al., 2022). CNN based models can also better capture the activity of other visual areas such as the primary visual cortex (V1), than standard LN models (Kindel et al., 2019; Cadena et al., 2017). While these models have been shown to reliably predict responses to visual stimuli having similar image statistics as the training set, it is unclear whether they will generalize to stimuli with different image statistics, such as different ambient luminance levels.

## 2.2 HOW THE VISUAL SYSTEM FUNCTIONS IN DYNAMIC AMBIENT LIGHT CONDITIONS

The light falling on our retina spans over twelve orders of magnitude and can change at different timescales: local changes on retina can occur within a few hundred milliseconds as we make eye movements such as saccades or by shadows cast on a tracked object whereas slower global changes can occur on a cloudy day. While extracting features under such dynamic conditions is still a challenge for artificial vision systems (Janai et al., 2020; Gomez-Ojeda et al., 2015; Kolaman et al., 2019), our retina effortlessly extracts reliable information for the downstream areas to make sense of our visual world (Tikidji-Hamburyan et al., 2015; Idrees et al., 2020; 2022; Goldin et al., 2022; Schreyer & Gollisch, 2021; Hosoya et al., 2005; Kastner & Baccus, 2011). This is achieved in part by signal processing done by photoreceptors (Angueyra et al., 2021; Yu et al., 2022; Clark et al., 2013), the very first cells in the visual processing hierarchy. They adapt strongly to match their neural sensitivity to the ambient light conditions. This adaptation itself is highly dynamic, and occurs at multiple timescales. As a result, the photoreceptor dynamics affects both the gain and the kinetics with which light inputs are converted into electrical signals in the retina. This adaptation is also highly local: each photoreceptor can adapt independently. Biophysical models of phototransduction in photoreceptors that translate light input into electrical signals are well established (Lamb & Jr, 1992; Rieke & Baylor, 1998; Invergo et al., 2014), and faithfully capture a range of responses. A more recent model by Angueyra et al. (2021) incorporates adaptive changes in the photoreceptor output in conditions where light falling on the retina changes by large magnitudes rapidly. We implemented this signalling cascade as a fully-trainable neural network layer, and used it as the first layer to deep CNNs to predict RGC responses to visual stimuli.

# 3 METHODS

## 3.1 PHOTORECEPTOR LAYER

Our new photoreceptor CNN layer builds upon a biophysical model of the phototransduction cascade by Angueyra et al. (2021) that incorporates the various feedforward and feedback molecular processes that convert photons into electrical signals, and therefore faithfully captures the photoreceptors' adaptation mechanisms (Fig. 1).

We implemented this biophysical model as a fully-trainable neural network layer in Keras (Chollet et al., 2015) in Python. This layer converts time-varying light intensity signal in units of receptor activations per photoreceptor per second (R*receptor$^{-1}$s$^{-1}$) into photocurrents (pA) by solving the differential equations that underlie the phototransduction cascade (biophysical model from Angueyra et al. (2021) reproduced in Appendix A). The photoreceptor layer has twelve parameters corresponding to the twelve interaction parameters of the biophysical model, all of which can (if needed) be trained through backpropagation using the Keras and TensorFlow package in Python. We fixed 8 parameters to their values from photore-

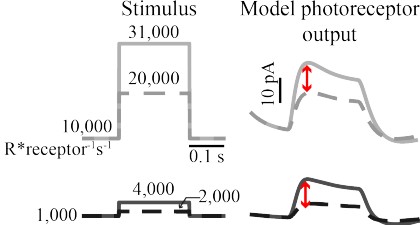

Figure 1: Dynamic gain adjustment in the photoreceptor model. Left column: step stimuli at two light levels: 10,000 R*receptor$^{-1}$s$^{-1}$(top) and 1,000 R*receptor$^{-1}$s$^{-1}$(bottom). Right column: model photoreceptor output to the stimuli on left. The difference in response to two small intensity steps at the lower light level (red arrow in bottom traces) was similar to two large intensity steps at a higher light level (red arrow in top traces), showing that the model photoreceptor dynamically adjusts its sensitivity (Angueyra et al., 2021).

ceptor data fits. These parameters relate to molecular processes such as the concentration of cyclic guanosine monophosphate (cGMP) in darkness and other factors governing cGMP conversion into current, and are similar for both rod and cone photoreceptors. The remaining 4 parameters, set to be trainable, include the rate and the gain factors of the molecular processes such as the opsin activation, the phosphodiesterase activation and decay, and the rate of calcium removal. These trainable parameters differ across photoreceptor types. In the current version of our model, the photoreceptor

parameters are shared by all the input pixels: I.e., the conversion of each pixel into photocurrents only depends on that pixel's previous values and not on the values of the other pixels.

## 3.2 MODEL ARCHITECTURE

We used our new photoreceptor layer at the input stage of a deep CNN (Fig. 2). The general architecture of our deep CNN was similar to Deep Retina (McIntosh et al., 2016), which we describe briefly here. In Deep Retina, the model input was a movie which passed through a convolutional layer operating in both space and time, followed by another convolutional layer that only operated in the spatial dimension. The output of the final dense layer was the predicted spike rate of RGCs. Our model differs from Deep Retina in two key ways: 1) the addition of the photoreceptor layer at the input stage; and 2) the use of batch normalization layers. We performed some experiments on models in which the photoreceptor layer was removed, and some in which the normalization layers were removed. These latter experiments provide the closest comparisons to Deep Retina.

Similar to Deep Retina, our model takes as input a movie (180 frames per training example where each frame corresponds to 8 ms) and outputs an instantaneous spike rate for each RGC at the end of that movie segment. To obtain the time series of RGC responses to longer movie stimuli, we feed into the model many 180-frame video samples taken from that longer movie, that correspond to 1-frame shifts. I.e., the model receives as inputs frames 1–180, 2–181, 3–182, etc., and outputs RGC responses at the times of movie frames 180, 181, 182, etc.

The photoreceptor layer of our model, converts each pixel of the input movie (where each pixel is in R*receptor$^{-1}$s$^{-1}$) into photocurrents (pA), based on the history of the pixel's value. The first 60 time points are then discarded to account for edge effects. The output of this layer is a movie that is 120 frames long, and the same spatial dimensions as the input visual stimuli. This photocurrents movie is then normalized by its mean and variance using a Layer Normalization layer (LayerNorm). The resulting movie is then passed through a conventional 3-layer CNN, where the first layer is a 3D convolutional layer operating in both the spatial and temporal dimensions. The output of the 3D convolutional layer is a 2D image which is normalized using Batch Normalization (BatchNorm) and then passed through a rectifying (ReLU) nonlinearity. To down sample the spatial dimensions, we applied a 2D max pool operation that took the maximum value over 2x2 patches of the previous layer's output. The subsequent 2D CNN layers are followed by a final, fully connected layer that outputs the predicted spike rate for each RGC in the dataset. We refer to this model as the photoreceptor-CNN model, or PR-CNN model for short.

Model parameters were optimized using Adam (Kingma & Ba, 2015), where the loss function was given by the negative log-likelihood under Poisson spike generation. Photoreceptor parameters were initialized to their known values from fits to experimental data. This initialization helped stabilize the joint training of the photoreceptor model with the downstream layers. The network layers were regularized with a $L_2$ weight penalty at each layer, including the photoreceptor layer, to prevent loss of information and be more robust to outliers. In addition, a $L_1$ penalty was applied to the output of the dense layer because the neural activity itself is relatively sparse and $L_1$ penalties are known to induce sparsity. The number of channels in each CNN layer and the filter sizes were optimized by a grid search for each model type. These details therefore differed for the models with and without the photoreceptor layer or normalization layers. Learning rates were set to 0.001.

## 3.3 RETINA ELECTROPHYSIOLOGY

To develop and test our models of the retina, we used data from ex vivo macaque monkey and rat retina electrophysiology experiments. The rat data was previously published by Ruda et al. (2020), whereas the monkey data were collected by us as part of this study. The basic experimental approach was the same in both sets of experiments: the retina of an euthanized animal was extracted and placed with the RGC-side down on a multi-electrode array (MEA).

We recorded monkey RGC activity to the same stimuli at three different mean light levels, each differing by 1 log unit: 0.3 R*receptor$^{-1}$s$^{-1}$, 3 R*receptor$^{-1}$s$^{-1}$ and 30 R*receptor$^{-1}$s$^{-1}$. These light levels fall under the scotopic regime, where mostly rod photoreceptors contribute to vision. The stimulus was a binary white noise checkerboard movie with 39 pixels x 30 pixels, where each pixel edge corresponded to approximately 140 $\mu$m on the retina surface. The movies across

the three light levels only differed in their mean pixel values which were 0.3 R*receptor$^{-1}$s$^{-1}$, 3 R*receptor$^{-1}$s$^{-1}$and 30 R*receptor$^{-1}$s$^{-1}$. In this work, we used a subset of 37 recorded RGCs that were classified as high quality units after spike sorting. This subset contained both the midget RGCs (that process fine details given their small receptive fields) and parasol RGCs (sensitive to motion given their large receptive fields) of ON and OFF types.

The rat experiments of Ruda et al. (2020) were performed at two light levels differing by 4 log units: 1 R*receptor$^{-1}$s$^{-1}$(scotopic light level where mostly rod photoreceptors contribute to vision) and 10,000 R*receptor$^{-1}$s$^{-1}$(photopic light level where cone photoreceptors predominantly contribute to vision). The white noise checkerboard movie in these experiments had 10 pixels x 11 pixels, with each pixel edge corresponding to approximately 252 $\mu$m on the retina. In this work, we used data from two rat experiments: a subset of 61 RGCs from Retina A and a subset of 58 RGCs from Retina B that were classified as high quality units after spike sorting. This subset contained OFF brisk sustained and OFF brisk transient RGC subtypes.

### 3.4 MODEL TRAINING AND EVALUATION

Models were trained using 40 minutes of neural recordings and associated stimulus movies for at least 50 epochs with early stopping based on a validation dataset separate from the test dataset used for measuring the model's performance.

Trained models were then evaluated using the held out test dataset that contained multiple repeats of 5-10 s of a binary white noise checkerboard movie not seen during the training, along with the associated neural recordings. We quantified the model performance with the fraction of explainable variance in each RGC's response that was explained by the model (FEV). This quantity (Eq. 1) was calculated as the ratio between the variance accounted for by the model and the *explainable* variance (denominator in Eq. 1). Such metrics to quantify how well a model predicts neural data have been used in previous studies like Cadena et al. (2017). We calculate FEV as

$$FEV = 1 - \frac{\frac{1}{T}\sum_{t=1}^{T}(y_t^A - \hat{y}_t)^2 - \sigma_{noise}^2}{Var[y^A] - \sigma_{noise}^2} \tag{1}$$

where,

$$\sigma_{noise}^2 = \mathbb{E}_t[(y_t^A - y_t^B)^2] \tag{2}$$

$y^A$ and $y^B$ are the observed spike rate of an RGC calculated as an average across set of repeats $A$ and set of repeats $B$ respectively. The sets $A$ and $B$ were obtained by randomly splitting the total number of repeats into two. $\hat{y}_t$ represents the predicted spike rate by the model at time bin $t$. The explainable variance (denominator in Eq. 1) is the variance of each RGC attributable to the stimulus, computed by subtracting an estimate of the observed noise from the variance across time (Eq. 3) in the actual RGC's responses, calculated as

$$Var[y^A] = \frac{1}{T}\sum_{t=1}^{T}(y_t^A - \bar{y}^A)^2 \tag{3}$$

where $\bar{y}^A$ is the the observed spike rate $y^A$ averaged across time. In all neural data sets we considered, the number of trials was large and hence the estimated noise variance was quite low. As a result, our FEV values are quite similar to what is obtained using the usual fraction explained variance calculation, which does not correct for unexplainable noise. By definition, FEV can be negative if the prediction error is larger than the variance in the actual responses. We report each model's performance across all RGCs as the median FEV across the set of RGCs. For ease in interpretation, we present FEV as a percentage throughout our results.

## 4 RESULTS

### 4.1 EXISTING DEEP LEARNING MODELS OF THE RETINA DO NOT GENERALIZE WELL ACROSS LIGHT LEVELS

We first tested whether existing deep learning models of the retina could generalize across light levels. To do this, we used the Deep Retina architecture (McIntosh et al., 2016) (i.e., the architecture

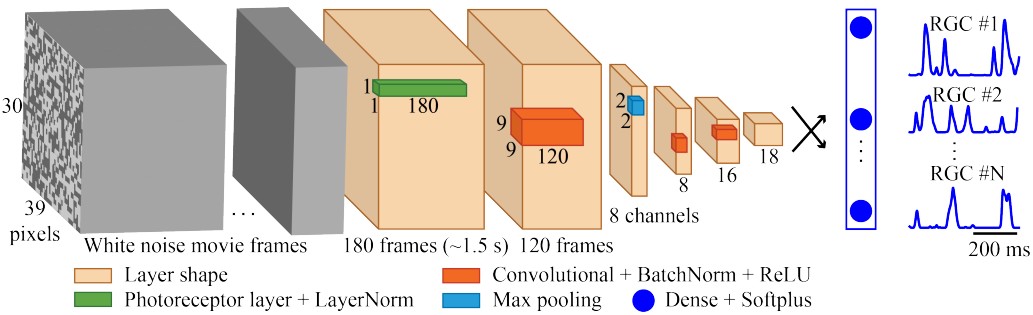

Figure 2: PR-CNN model architecture. The photoreceptor layer at the input takes 180 frames of a movie as a single sample and converts each pixel (green color) into photocurrents. The photocurrents are then passed to the first convolution layer that has 8 channels where each kernel (orange color) operates over spatial dimensions of 9 pixel x 9 pixel and temporal depth of 120 frames. The result is normalized through a Batch Normalization layer and then passes through a Rectified Linear Unit (ReLU). The output is then max pooled to reduce the spatial dimension and then passes through another two convolution layers. The model output layer is a dense layer that has N units based on the number of RGCs in the dataset. It converts previous layer outputs into RGC spiking output (blue traces). By traversing through the input movie 180 frames at a time, an entire time series of RGC responses is achieved.

in Fig. 2 with the photoreceptor layer and normalization layers removed), and re-optimized the numbers of convolutional layers, filters in each layer, and the filter sizes with a grid search. This model had three CNN layers followed by a dense layer with 8, 16 and 18 channels in each of the CNN layers, respectively, and filter sizes of 9x9, 7x7 and 5x5 respectively.

We trained this model to predict monkey RGC responses using the training dataset that contained binary white noise movies at 30 R*receptor$^{-1}$s$^{-1}$and 3 R*receptor$^{-1}$s$^{-1}$, the two higher light levels in the monkey data set. We then used the test data set to evaluate its performance at all light levels including the testing light level (0.3 R*receptor$^{-1}$s$^{-1}$) not used during the training (Fig. 3a). A Layer Normalization layer at the input normalized each pixel in input movie sample by the mean and variance across all pixels in the movie. The model could reliably predict responses to stimuli at the two training light levels, 30 R*receptor$^{-1}$s$^{-1}$and 3 R*receptor$^{-1}$s$^{-1}$(Fig. 3b, columns 1-2), with FEV ($median \pm 95\% c.i.$) of 84% ± 11% and 78% ± 3% respectively (Fig. 4a). However, this model performed poorly at the testing light level (0.3 R*receptor$^{-1}$s$^{-1}$) (Fig. 3b, column 3) explaining only 16% ± 15% of the variance in the population data (Fig. 4a).

To overcome internal covariate shift and stabilize the network, we included batch normalization layers prior to each activation function in the model. This marginally increased the overall performance of the model at each light level (Fig. 4b). The generalization performance at the test light level was however still quite low with FEV of 24% ± 15% (Fig. 4b).

## 4.2 PHOTORECEPTOR FRONT END ENABLES DEEP LEARNING MODELS OF RETINA TO GENERALIZE ACROSS LIGHT LEVELS

To test whether our new PR-CNN model was better at generalizing across lighting conditions, we repeated the same task as above, i.e., trained the model end-to-end including the photoreceptor layer to predict monkey RGC responses to the binary white noise movie at the two higher light levels, 30 R*receptor$^{-1}$s$^{-1}$and 3 R*receptor$^{-1}$s$^{-1}$. Similar to the model without the photoreceptor layer, this model could also reliably predict responses to test stimuli at the two training light levels (Fig. 4c columns 1-2 and Fig. 4c). Importantly, this model was able to explain 54% ± 11% of the variance in responses at a light level lower than those at which it was trained (Fig. 4c column 3 and Fig. 4c). This is a notable improvement over the model without the photoreceptor layer (but with the normalization layers included), which only explained 24% ± 15% of the variance (Fig. 4b).

Next, we repeated this analysis with all different combinations of training and testing light levels: each time, we trained the model on data from two light levels, and then evaluated the model on test

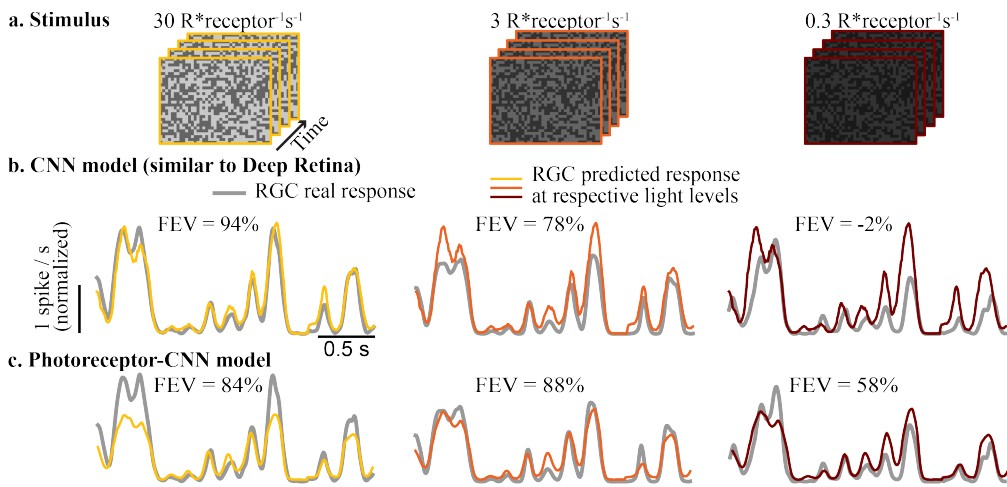

Figure 3: Example monkey retinal ganglion cell (RGC). **a.** White noise movie at three different light levels (columns). Models were trained on data at 30 R*receptor$^{-1}$s$^{-1}$(column 1) and 3 R*receptor$^{-1}$s$^{-1}$(column 2). **b.** Actual spike rate of an example RGC (gray) and those predicted by a CNN model similar to Deep Retina (colored) at each light level in **a** (columns). FEV values above each trace quantifies the performance of the model for this RGC at the corresponding light levels. **c.** Same as in **a** but for our photoreceptor-CNN model.

data from a different light level (Fig. 4d). For all combinations of training and testing light levels, the PR-CNN model performed better at generalizing to new light levels than the conventional CNN model without the photoreceptor layer (but with normalization layers included). The difference was smallest for the case where the testing light level (3 R*receptor$^{-1}$s$^{-1}$) was intermediate between the training light levels (30 R*receptor$^{-1}$s$^{-1}$and 0.3 R*receptor$^{-1}$s$^{-1}$) (Fig. 4d, column 3). This suggests that while standard CNN based models can generalize to conditions that can be interpolated from training conditions, they are not as well equipped to generalize to conditions that have to be extrapolated. Our new PR-CNN model leverages the biophysics of photoreceptor adaptation to do much better at this challenging extrapolation task.

### 4.3 GENERALIZATION ACROSS EXTREMELY DIFFERENT LIGHT LEVELS

Having observed that our PR-CNN model can generalize well between light levels that differ by 1-2 orders of magnitude, we next wondered whether it could also generalize well across more extreme variations in lighting. To answer this question, we trained our model to predict responses at a relatively bright (photopic) light level (10,000 R*receptor$^{-1}$s$^{-1}$) where cone photoreceptors predominantly contribute to vision and evaluated the model's generalization performance at a much dimmer (scotopic) light level (1 R*receptor$^{-1}$s$^{-1}$) where only rod photoreceptors are active.

For comparison, we first performed this experiment with the conventional CNN model including normalization layers (as in Fig. 4b). That model failed (FEV of $-52\% \pm 9\%$) to predict responses at the 1 R*receptor$^{-1}$s$^{-1}$light level after having been trained at the 10,000 R*receptor$^{-1}$s$^{-1}$level (Fig. 5a,c). Our proposed PR-CNN model, however, did notably well (Fig. 5b), achieving FEV of $54\% \pm 8\%$ on this task (Fig. 5). For this experiment, we trained a PR-CNN model at 10,000 R*receptor$^{-1}$s$^{-1}$and then replaced that model's photoreceptor parameters (which correspond to cone cells at this light level) with those corresponding to rod cells, as explained below and in Fig. B.1 in Appendix B.

First, we trained our model end-to-end to predict rat RGC responses (from Retina A) to stimuli at the bright (photopic) light level (10,000 R*receptor$^{-1}$s$^{-1}$). This led to an estimate of photoreceptor and CNN parameters for the cone pathway. We then estimated parameters reflecting rod photoreceptors by re-training the model at the dim (scotopic) light level (1 R*receptor$^{-1}$s$^{-1}$), while keeping the CNN weights and biases fixed to their already-learnt values. Thus, the retraining altered only the photoreceptor parameters based on the data from Retina A.

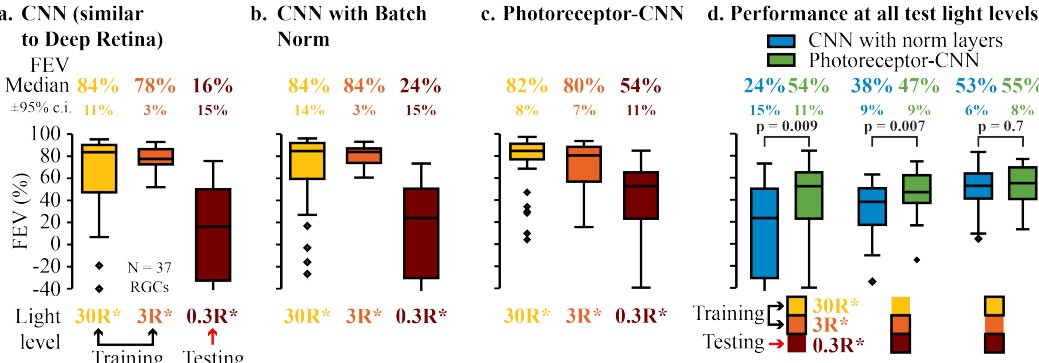

Figure 4: Model performance comparisons. Performance on the test data set for **(a)** a CNN model that had a similar architecture as Deep Retina, **(b)** CNN model of **a** but with normalization layers included, and **(c)** our proposed photoreceptor-CNN model. **a-c.** Each model was evaluated at three light levels (labelled below each box plot): 30 R*receptor$^{-1}$s$^{-1}$and 3 R*receptor$^{-1}$s$^{-1}$at which the models were trained at and 0.3R*receptor$^{-1}$s$^{-1}$(testing condition) which the models did not see during the training. The box plot at each light level shows the distribution of FEVs across 37 monkey RGCs. Numbers at the top of each box plot are the median FEVs $\pm$ 95%$c.i.$. **d.** Performance of the CNN model with normalization layers (blue color; same model as in **b**), and the photoreceptor-CNN model (green color; same model as in **c**) at all combinations of training and test light levels. For each column, the legend below the box plot panel shows the two light levels the models were trained at (black outline) and the third light level at which it was tested. The box plots show the distribution of FEVs at this testing light level. Testing light levels were 0.3 R*receptor$^{-1}$s$^{-1}$(column 1), 3 R*receptor$^{-1}$s$^{-1}$(column 2), and 30 R*receptor$^{-1}$s$^{-1}$(column 3). p-values were calculated by performing two-sample t-test on the FEV distributions from the CNN and PR-CNN model at each testing light level.

We then trained the PR-CNN model to predict RGC responses of another rat's retina (Retina B) to stimuli at the bright light level. Here, we fixed the photoreceptor parameters to the values learned from Retina A at the bright light level. We then exchanged those photoreceptor parameters with the ones learned from Retina A at the low light level. In this manner, we took the model where the CNN weights were learned on model of Retina B at the bright light level, and simply exchanged the cone photoreceptor parameters for rod parameters. Notably, in this procedure, none of the data from Retina B at the dim light level was used in training the model.

In retrospect, models without inherent knowledge of light levels, like the conventional CNN, are likely to fail at this generalization task as response properties of RGCs are substantially different at bright (photopic) and dim (scotopic) light levels. For example, rod mediated RGC responses at dim conditions are slower than cone mediated responses at brighter light levels (Ruda et al., 2020; Baylor & Fettiplace, 1977). This could explain the temporal lag between the predicted and actual response at dim light level (Fig. 5a): the model trained at bright light levels (10,000 R*receptor$^{-1}$s$^{-1}$) learnt faster kinetics of the cone pathway. While some of the differences in RGC responses may arise due to faster response kinetics of the cones themselves (Cao et al., 2007; Baylor & Hodgkin, 1973; Ingram et al., 2016; Schneeweis & Schnapf, 1995), the relative contribution of photoreceptors and downstream retinal circuits are not well understood. Our PR-CNN model could be one way to dissect such underlying contributions.

## 5   DISCUSSION

Here, we introduced a new class of machine learning models for neuroscience applications. The input layer to the model is a novel convolutional layer that incorporates a biophysical photoreceptor model to leverage photoreceptor adaptation mechanisms and enable generalization across light levels. We found that this model strongly outperforms other CNN architectures, in generalizing across light levels, making it a potentially desirable tool for many applications, including visual prosthetics.

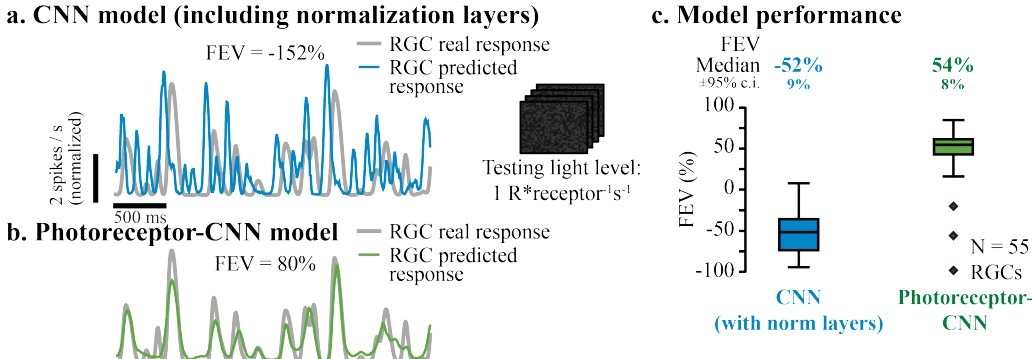

Figure 5: Generalization across extreme light levels. **a-b.** Rat Retina B example RGC responses at scotopic light level (1 R*receptor$^{-1}$s$^{-1}$). Actual responses shown in gray. Predicted responses generated by the CNN model (trained at photopic level) without the photoreceptor layer (**a**, blue) and with a photoreceptor layer(**b**, green). **c.** Box plots show the distribution of FEVs across RGCs (N = 55 RGCs, Retina B) for the CNN model with normalization layers (blue) and for the CNN model with the photoreceptor layer added, when the models were trained at photopic light level and tested at scotopic light level.

The photoreceptor model while complex, has parameters that map directly onto the biology, presenting the scientific community with an opportunity to investigate visual processing of dynamic stimuli. For example, how much of the adaptation in the retina can be accounted for by photoreceptors alone? And how much does the sensitivity of RGCs to different stimulus features vary with lighting conditions? Moreover, models of retina that can leverage the power of deep learning to model multiple ganglion cells simultaneously, together with biologically interpretable components, could be used to dissect the relative contributions by different components. From a wider neuroscience perspective, our approach demonstrates the power of embedding neural dynamics in deep learning models of the brain: biophysical layers can match the sensitivity to prevailing input conditions that change dynamically whereas the downstream layers of the model can progressively extract relevant features from those dynamically adapting input stages.

The wider scientific community might not be interested in biological investigations, but could still benefit from the photoreceptor's light adaptation mechanisms to overcome challenges in computer vision. To this end we have developed an adaptive convolutional layer based upon a simplified phenomenological photoreceptor model (Appendix C.1). This model could potentially enable better computer vision systems for applications where lighting is highly dynamic, such as the object detection and recognition systems on autonomous vehicles. Here, pixel-wise local luminance adaptation (from using our adaptive-convolutional layer as an input layer) could help filter out disruptive lighting changes induced by changing shadows, that would otherwise cause errors in object recognition and/or detection (Janai et al., 2020; Gomez-Ojeda et al., 2015; Kolaman et al., 2019). A proof-of-concept that a photoreceptor-inspired convolution layer, Adaptive-Conv, may be better than conventional CNNs at this task, is presented in Appendix C.1.

Future work includes overcoming some of the limitations of this work. For example, the manual switch from cone to rod photoreceptors that is required for a model trained at cone light levels to generalize to rod light levels (Fig. 5). In addition, the photoreceptor parameters learnt may also reflect some of the adaptive mechanisms in downstream retinal circuits. While this is not a concern when developing models that perform better at different light levels, it may limit some biological inferences. One way to overcome this limitation would be by adding layers downstream of the photoreceptor layer that incorporate other known biological sources of adaptation in the retina.

## REPRODUCIBILITY STATEMENT

The photoreceptor-CNN model and Adaptive-Conv model can be downloaded from [Redacted for Double-Blind Review]. All retina electrophysiology data used in this study can be made available upon reasonable request.

ETHICS STATEMENT

All electrophysiology experiments were performed in accordance with the guidelines of Institutional Animal Care and Use Committee at [Redacted for Double-Blind Review].

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

## A    APPENDIX: PHOTORECEPTOR BIOPHYSICAL MODEL

The biophysical model of the photoreceptor that we used is described in Angueyra et al. (2021). Below we reproduce the model to facilitate the readers.

Input to the model is the stimulus intensity as a function of time (in units of R*receptor$^{-1}$s$^{-1}$) given by $Stim(t)$ in Eq. 4. The output of the model is the photoreceptor outer segment current as a function of time, $I(t)$ given in Eq. 11.

$$\frac{dR^*(t)}{dt} = \gamma Stim(t) - \sigma R^*(t) \tag{4}$$

$$\frac{dP(t)}{dt} = R^*(t) - \phi P(t) + \eta \tag{5}$$

$$\frac{dG(t)}{dt} = S(t) - P(t)G(t) \tag{6}$$

$$\frac{dCa(t)}{dt} = qI(t) - \beta Ca(t) \tag{7}$$

$$Ca(0) = C_{dark} \tag{8}$$

$$S(t) = \frac{S_{max}}{1 + \frac{Ca(t)}{K_{GC}}^m} \tag{9}$$

$$\frac{dCa_{slow}(t)}{dt} = \beta_{slow}(Ca_{slow}(t) - Ca(t)) \tag{10}$$

$$I(t) = k_{Ca}G(t)^h \tag{11}$$

In our Keras photoreceptor layer, the parameters $\sigma$, $\gamma$, $\phi$, $\eta$, $q$, $\beta$, $C_{dark}$, $K_{GC}$, $B_{slow}$, $m$, $k_{Ca}$, and $h$ can all be set to trainable. In this study however, we only allowed $\sigma$, $\beta$, $\phi$ and $\eta$ to be trained. Remaining parameters were fixed to their values (Angueyra et al., 2021) from photoreceptor data fits.

## B APPENDIX: TRAINING PROCEDURE FOR GENERALIZING ACROSS EXTREME LIGHT LEVELS

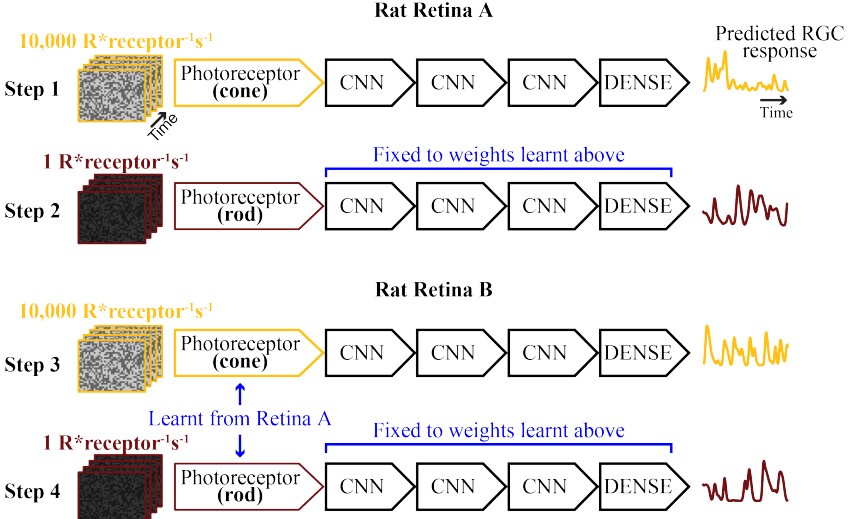

Figure B.1: Schematic for training across extreme light levels. **Step 1:** PR-CNN model was trained end-to-end to predict Rat Retina 'A' RGC responses at photopic light level (10,000 R*receptor$^{-1}$s$^{-1}$). This led to an estimate for cone photoreceptor parameters and a model for the inner retina circuit (the CNN layers). **Step 2:** The model was re-trained at scotopic light level (1 R*receptor$^{-1}$s$^{-1}$) but the CNN layers were set to non-trainable and fixed to weights learnt in Step 1. In this case, the photoreceptor model learnt parameters reflecting rods. **Step 3:** The model was trained to predict Rat Retina 'B' responses at photopic light level. Photoreceptor layer was set to non-trainable with parameters fixed to cone parameters learnt in **Step 1**. **Step 4 (testing step):** Model was tested to predict Rat Retina 'B' responses at scotopic light level. Here, we used the rod photoreceptor parameters learnt in **Step 2** and rest of the model representing inner retina pathways of Retina B learnt in **Step 3**.

## C APPENDIX: ADAPTIVE-CONV

### C.1 PROOF-OF-CONCEPT DEMONSTRATION OF APPLICATION IN COMPUTER VISION

Given that the photoreceptor adaptive mechanisms enable the retina (and the PR-CNN model discussed in the main paper) to work under a wide range of steady-state and dynamic light intensities, we hypothesized that those same adaptive mechanisms could also be useful for computer vision tasks in dynamic lighting conditions. Specifically, we consider conditions with potential disruptive events such as shifting cloud cover or shadows due to moving tree branches, that can rapidly and dramatically change the lighting in part or all of a visual scene. These events can shift the scale of the pixel values in an image, either on some pixels or on all of them, depending on the spatial scale of the event.

If the spatial scale is large, and the whole image exhibits the change in lighting level, simple normalization (i.e., layer norm) could remove the change. For localized changes, however, there is no obvious strategy through which normalization can remove the lighting change. Moreover, these local changes in pixel intensities can be easily confused for changes in the edges in the scene, hence confounding object detection and identification (Gomez-Ojeda et al., 2015; Kolaman et al., 2019; Janai et al., 2020).

To address this challenge, we have developed a novel convolution layer, the Adaptive-Conv, which contains a simplified functional model of the photoreceptors (see Appendix. C.2 for technical details). As a proof-of-concept that such photoreceptor-inspired models could be better at solving

computer vision tasks, we considered a toy example where the input to the model is an object's intensity represented by a single pixel in space. The pixel value (intensity) varies in time around a fixed mean value (object intensity in Fig. C.1a, column 1). At different time points, brief disruptive events of different durations and intensities (spanning 1 to 5 log units) (source intensity in Fig. C.1a, column 2), change the amount of light reflected off the object( Fig. C.1a, column 3). We attempted to train neural networks with, or without, our new Adaptive-Conv layer, to extract the original intensity of the object (red line in Fig. C.1a, column 3), thereby removing the effects of the disruptive light change (blue line in Fig. C.1a, column 3)

We trained a model with the Adaptive-Conv layer to perform this task (Fig. C.1b, left panel). The model takes as input a movie with 200 frames. The first layer is an Adaptive-Conv layer with 40 channels followed by 7 dense layers each containing 80 units. These hyperparameters were optimized using a grid search. The width of the temporal kernel in the Adaptive-Conv layer was set to 200. This model could reliably extract the original intensity of the object, discounting the disruptive events (Fig. C.1b, right panel). In contrast, a conventional 1D CNN model where the standard convolution layer was used (Fig. C.1c, left panel), failed to extract the original intensity following the disruptive events (Fig. C.1c, right panel), despite having 4 times as many parameters as the model with the Adaptive-Conv. (For this model as well we optimized the hyperparameters using a grid search which resulted in 100 channels for the first convolution layer, followed by 5 dense layers each with 200 units.) Our ongoing work will test this Adaptive-Conv approach on more complex vision tasks and see if the same observation holds, namely that the photoreceptor model discounts the sudden large changes in intensity, thereby removing the impacts of those intensity changes on object detection and/or categorization.

## C.2 ADAPTIVE-CONV LAYER

The Adaptive-Conv layer is a trainable layer that we constructed using Keras (Chollet et al., 2015) package in Python and builds upon a modified version of a phenomenological photoreceptor model described in (Clark et al., 2013). In this layer, the input signal at each pixel, $x(t)$ is convolved in time, $t$, with two temporal kernels, as in

$$y(t) = \int_{-\infty}^{t} K_y(t - t')x(t')dt' \tag{12}$$

$$z(t) = \int_{-\infty}^{t} K_z(t - t')x(t')dt', \tag{13}$$

where $x(t)$ is the pixel intensity at time $t$, $K_y$ and $K_z$ are the two temporal, of the form

$$K_y(t) = \frac{t^{n_y}}{n_y! \tau_y^{n_y+1}} e^{\frac{-t}{\tau_y}} \theta(t) \tag{14}$$

$$K_c(t) = \frac{t^{n_c}}{n_c! \tau_c^{n_c+1}} e^{\frac{-t}{\tau_c}} \theta(t) \tag{15}$$

$$K_z(t) = \gamma K_c(t) + (1 - \gamma)\frac{t^{n_z}}{n_z! \tau_z^{n_z+1}} e^{\frac{-t}{\tau_z}} \theta(t) \tag{16}$$

Here, $\tau_y$, $\tau_c$, $\tau_z$, specifies the time scale and $n_y$, $n_c$, $n_z$ specifies the rise behaviour of the kernels. Parameter $\gamma$ (constrained at $0 \leq \gamma \leq 1$) weights the relative importance of the two kernels with different temporal profiles making up $K_z$, and $\theta$ is the Heavyside function. All these parameters are trainable within the Adaptive-Conv layer.

The two convolved input signals, $y(t)$ and $z(t)$ are then weighted and combined divisively to form the Adaptive-Conv layer's output, through the following operation

$$o(t) = \frac{\alpha}{\kappa + \beta z(t)} y(t), \tag{17}$$

where $\alpha$, $\kappa$ and $\gamma$ are trainable weights. In total, the Adaptive-Conv layer has 10 parameters, all of which are fully trainable through backpropagation.

Similar to the biophysical model used in the main paper, a key feature of this phenomenological model is that the gain of the input is dynamically controlled by $\frac{\alpha}{\kappa + \beta z(t)}$ based on prevailing inputs over the time scale of $K_z$.

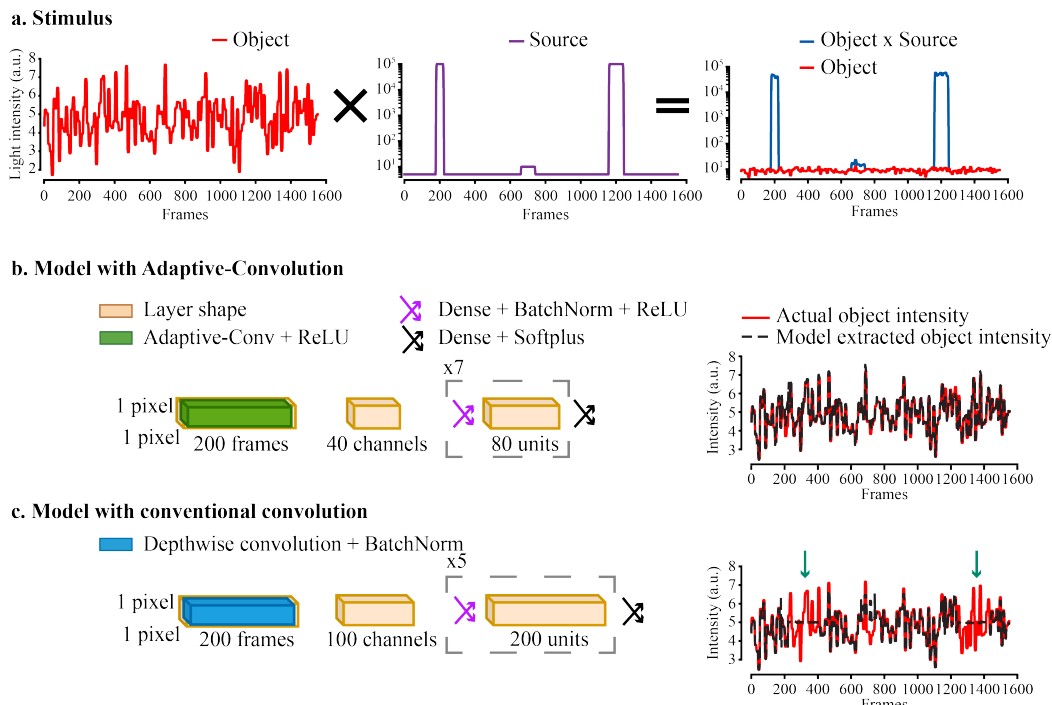

Figure C.1: Adaptive-Conv layer. **a.** Snippet of the toy example stimuli where intensity reflected off an object varies in time around a mean value (red trace in column 1) is multiplied by a signal having large intensity variations (log scale; purple trace in column 2) to introduce disruptions in the intensity reflected off the object (blue trace in column 3). **b.** Left panel: a model with the Adaptive-Conv layer trained to extract the original intensity of the object from the combined signal (blue in **a**). Right panel: original intensity of the object that the model is trained to extract (red solid line) and the intensity extracted by the model (dashed black line). **c.** Same as in **b** but with a model where a conventional convolution layer replaced our Adaptive-Conv layer. Cyan arrows in the right panel indicate examples where the model with conventional convolution layer failed to extract the object intensity from the combined signal.

