# OpenReview forum: "A new photoreceptor-inspired CNN layer enables deep learning models of retina to generalize across lighting conditions"
_ICLR.cc/2023/Conference — Submitted to ICLR 2023_

### Official Review · Reviewer_Rcws · 2022-10-21

**Confidence:** 3
**Correctness:** 2
**Technical Novelty And Significance:** 3
**Empirical Novelty And Significance:** 2
**Recommendation:** 3

**Clarity, Quality, Novelty And Reproducibility:**

The paper is well written. It is an interesting, though somewhat incomplete effort at biologically plausible computer vision. It is not clear whether data and source code will be provided to make this reproducible.

**Details Of Ethics Concerns:**

Experiments were done on macaque monkeys and rats. AC should make sure there are no conflicts with ICLR guidelines

**Strength And Weaknesses:**

Strengths:

S1: the authors identify and important problem in computer vision algorithms and make some biologically motivated suggestions on how this problem could be solved.

S2: the paper is extremely well written and is up to ICLR standards

Weaknesses:

W1: With respect to making computer vision algorithms more robust, why do they feel that such an adaptation approach is necessary? Why not simply give HDR data (high dynamic range data) as input to algorithms (ie data that is represented with more than 8 bits per channel). Wouldn't this be sufficient to deal with saturation issues in high intensity situations, and low contrast in low light situations?

W2: page 1 typo: "We focus here on retina". Missing "the"

W3: prior related work discussed in the paper does not go before 2015-2016. I suggest an expansion on prior work detailing the effects of sensor bias in experimental methods.

W4: section 3.3. The authors do electrophysiology experiments from ex vivo macaque monkey and rat retina. It is not often that you see such experiments in machine learning/computer vision conferences. Machine learning conferences like ICLR are more focused on algorithms that improve real-world performance. The way the paper is written it seems to be focused more towards a computational neuroscience audience. While not necessarily bad, in this case this seems to be to the detriment of demonstrating whether this algorithm actually leads to improvements in accuracy, performance, speed, power of computer vision systems. The main focus of the authors should have been to show that their algorithm leads to some quantifiable improvements compared to other algorithms that are solving a certain real-world problem. The authors seem to try to touch upon this in the appendix, but not in a lot of detail. This is the paper's rather major weakness in my opinion. The appendix algorithm should be in the main paper and its evaluation on a real world problem expanded upon.

W5: page 5: I suggest the authors introduce parasol and midget cells and how their behavior differs, since the typical ICLR audience will not be familiar.

W6: Eq 1: Should it be T or T-1? Is it sample of population variance?

W7: Eq 1: clarify how Var[y] was obtained.

W8: page 5: how were the sets A, B obtained? Are there confidence intervals on these estimators? Is eq 2 an unbiased estimator? Clarify if this estimator is used in Cadena 2017. It is unclear whether the average is across time. Clarify over what the expected value is taken.

W9: will source code and the data be provided to make this reproducible?

W10: if you connect a live camera as input, how well would the algorithm adapt to changes in the sensor gain and shutter speed?

W11: page 6: 180 frames are used. At 30fps this is 6 seconds. This means it takes 6 seconds for adaptation to happen? Could this be a drawback in computer vision algorithms?

**Summary Of The Paper:**

The authors indicate that modern computer vision systems tend to fail under dynamic lighting conditions. This is because computer vision algorithms do not explicitly make use of adaptation mechanisms similar to the ones used in retinal ganglion cells. Because current deep learning models of the retina have no in-built notion of light level , they are unable to accurately predict RGC responses under lighting conditions that they were not trained on. The authors use a model of RGC and test it on monkey and rat retinal data. The authors argue that their algorithm can be used with computer vision algorithms to adapt to dynamic environments.

**Summary Of The Review:**

Overall, while this is a well written paper, I feel the authors need to do a better job adapting it towards the typical ICLR audience. Currently it is addressed more towards a computational neuroscience audience. As indicated on page 6, I do not feel comfortable accepting to ICLR a paper that has only been tested with white noise input. The effort in appendix C to compare it with actual algorithmic performance is incomplete and needs more work

---

> ### Author Response · Authors · 2022-11-17
> **Response 1/2**
>
> We thank the reviewer for their insightful comments on the strengths and weaknesses of the paper. We have updated the manuscript to overcome the weaknesses.
>
> **W1: With respect to making computer vision algorithms more robust, why do they feel that such an adaptation approach is necessary? Why not simply give HDR data (high dynamic range data) as input to algorithms (ie data that is represented with more than 8 bits per channel). Wouldn't this be sufficient to deal with saturation issues in high intensity situations, and low contrast in low light situations?**
>
> High dynamic range is definitely useful as it would allow image acquisition devices to capture a scene with a high luminance range and therefore avoid saturation. However for certain applications like object recognition, the problem is not only dealing with saturation but also dealing with changes in feature representation when parts of an object experience large intensity changes. One way to overcome this challenge is to find object representations that are invariant to local luminance fluctuations which is an active area of research. Another way could be to discount the local luminance fluctuations. The latter is what the photoreceptors partially accomplish and this can be leveraged towards computer vision applications.
>
> **W2: page 1 typo: "We focus here on retina". Missing "the"**
>
> Thanks for pointing this out. We have fixed this typo.
>
> **W3: prior related work discussed in the paper does not go before 2015-2016. I suggest an expansion on prior work detailing the effects of sensor bias in experimental methods.**
>
> Thank you for highlighting this area for literature search. This would be necessary if the primary focus of the paper was computer vision. We will take this into account in our upcoming work on developing the photoreceptor model for computer vision applications.
>
> **W4: section 3.3. The authors do electrophysiology experiments from ex vivo macaque monkey and rat retina. It is not often that you see such experiments in machine learning/computer vision conferences. Machine learning conferences like ICLR are more focused on algorithms that improve real-world performance. The way the paper is written it seems to be focused more towards a computational neuroscience audience. While not necessarily bad, in this case this seems to be to the detriment of demonstrating whether this algorithm actually leads to improvements in accuracy, performance, speed, power of computer vision systems. The main focus of the authors should have been to show that their algorithm leads to some quantifiable improvements compared to other algorithms that are solving a certain real-world problem. The authors seem to try to touch upon this in the appendix, but not in a lot of detail. This is the paper's rather major weakness in my opinion. The appendix algorithm should be in the main paper and its evaluation on a real world problem expanded upon.**
>
> From a neuroscience application perspective, predicting neural responses to stimulus in dynamic light conditions is as much of a real-world problem as recognizing objects under varying conditions of illumination. In fact, the neuroscience problem is more challenging because the target labels change in complicated ways as the lighting conditions change, whereas in the object recognition application, the target labels do not change. We demonstrate in the paper that our new ML model shows quantifiable improvement over existing state-of-the-art ML models of retina in predicting neural responses across light conditions. We acknowledge this is an ML application in neuroscience and hence the paper was submitted in the ICLR area of “Neuroscience and Cognitive Science”
>
> **W5: page 5: I suggest the authors introduce parasol and midget cells and how their behavior differs, since the typical ICLR audience will not be familiar.**
>
> We have updated the text to include this.
>
> **W6: Eq 1: Should it be T or T-1? Is it sample of population variance?**
>
> It is T, the population variance.
>
> **W7: Eq 1: clarify how Var[y] was obtained.**
>
> We have updated the paper to include this (Eq. 3). In short, y is the response averaged across half the total number of trials. It is the same as ytA. Var[y] is the population variance across time i.e. across time bins t.

---

> > ### Comment · Reviewer_Rcws · 2022-11-25
> > **my final response to the rebuttal**
> >
> > After reading the comments by the reviewers, authors and area chair I have decided to keep my ranking as is. Reviewer b3Ty has expressed the same concerns that I have regarding applications of this algorithm to vision. As such I encourage the authors to make use of this constructive criticism to try to improve the quality of the paper by better finalizing their experiments

---

> ### Author Response · Authors · 2022-11-17
> **Response 2/2**
>
> **W8: page 5: how were the sets A, B obtained? Are there confidence intervals on these estimators? Is eq 2 an unbiased estimator? Clarify if this estimator is used in Cadena 2017. It is unclear whether the average is across time. Clarify over what the expected value is taken.**
>
> Sets A and B were obtained by randomly splitting the total number of trials into two. There are no confidence intervals on these estimators. One can boot-strap and have these but given the large number of trials the estimated noise variance was quite low. We used a different noise estimator as the one used in Cadena 2017 given the differences in the datasets, specifically that our dataset contained movies and not static images. The expectation is taken over time.
>
> **W9: will source code and the data be provided to make this reproducible?**
>
> Source code will be provided freely via github. Data can be provided upon request: some parts of it are part of a larger project that precludes posting the data on the internet for unrestricted access.
>
> **W10: if you connect a live camera as input, how well would the algorithm adapt to changes in the sensor gain and shutter speed?**
>
> It would treat these changes as if the changes were a result of source intensity changes, because in its current form the algorithm does not take into account sensor gain and shutter speed changes. In terms of modeling the retina, the change in sensor gain (and dynamics) is precisely what our photoreceptor model captures.
>
> **W11: page 6: 180 frames are used. At 30fps this is 6 seconds. This means it takes 6 seconds for adaptation to happen? Could this be a drawback in computer vision algorithms?**
>
> In the neural datasets used, each image frame was shown on to the retina for ~8 ms, which corresponds to 125 fps. 180 frames is thus ~1.4 seconds. Retinal adaptation occurs at multiple timescales with fast adaptation occurring within tens of milliseconds and slow adaptation occurring over tens of seconds. For a computer vision application, the initial parameters of the model can be set to represent faster kinetics. In the demonstration of the model for the computer vision task, we used arbitrary units with a temporal window of 200 frames. Considering 1 ms time bin, it would be able to capture adaptation occurring within <200 ms.
>
> **The paper is well written. It is an interesting, though somewhat incomplete effort at biologically plausible computer vision. It is not clear whether data and source code will be provided to make this reproducible.**
>
> Thank you for the encouraging remarks regarding the quality of the paper. We would like to highlight that our paper is not an effort at biologically plausible computer vision. If so, we would have submitted the paper under a different area that encompasses computer vision. Our paper was submitted under the area of “Neuroscience and Cognitive Science” which better reflects the subject of our paper which is an ML application in neuroscience. We do mention that the source code for the models will be provided. The underlying data is being used in other neuroscience projects, but we will make it available upon request to anyone who wishes to experiment with it further or reproduce our results.
>
>
> **Overall, while this is a well written paper, I feel the authors need to do a better job adapting it towards the typical ICLR audience. Currently it is addressed more towards a computational neuroscience audience. As indicated on page 6, I do not feel comfortable accepting to ICLR a paper that has only been tested with white noise input. The effort in appendix C to compare it with actual algorithmic performance is incomplete and needs more work**
>
> We acknowledge this is an ML application in neuroscience and hence the paper was submitted in the ICLR area of “Neuroscience and Cognitive Science”. We understand the limitation that the model has only been tested with white noise input. However, existing data available at multiple light levels is limited to white noise. To match the stimuli to this available data, we also performed our experiments using white noise stimuli. Such stimuli also allows us to calculate spatio-temporal features allowing us to investigate how such features change across space. Using more naturalistic stimuli is indeed the goal and we will be conducting those experiments in the future as we work on developing this approach further.

---

> ### Comment · Reviewer_Rcws · 2022-11-22
> **response to author comments**
>
> Overall the authors have addressed all my questions in the rebuttal. As I indicated in my review, my main concern is that I feel that the experimental section of the paper does not really demonstrate that this algorithm leads to better performance in a real-world computer vision scenario, that would be of interest to a more engineering-oriented audience. If the paper was submitted under a more clear cut computer vision area, or in an engineering oriented conference like CVPR I would classify it as a definite reject. However as the authors indicate, they have labelled their paper as being part of the "Neuroscience and Cognitive Science" area. If the Area Chair indicates that neuroscience-related submissions to ICLR do not necessarily need to have an experimental section demonstrating some kind of improved performance in a real-world scenario, I am willing to modify the paper's ranking to an accept.

---

### Official Review · Reviewer_TU1Q · 2022-10-24

**Confidence:** 4
**Clarity, Quality, Novelty And Reproducibility:** Clear enough.
**Correctness:** 4
**Technical Novelty And Significance:** 3
**Empirical Novelty And Significance:** 4
**Recommendation:** 6

**Strength And Weaknesses:**

Introducing an adaptive gain control mechanism as a front end to CNN is a sensible and good idea.  The fact that such an adaptive mechanism should improve a CNN's prediction of neural response in different lighting conditions is not surprising. Nevertheless,  demonstrating empirically it actually does the job is still meaningful and an accomplishment. Making the adaptive mechanism itself partially learnable is also novel.

**Summary Of The Paper:**

The authors showed that putting a photoreceptor front-end with adaptive dynamic gain control allows a deep neural network to predict more reliably the responses of retinal ganglion cells in different lighting conditions than the STOA CNN retinal model.

**Summary Of The Review:**

While the technical and conceptual contribution in ML and representational learning are relatively limited, this paper is acceptable for demonstrating or proving  that an adaptive gain control mechanism as a front-end of CNN can improve the prediction of RGC's responses in different lighting condition. Even though it might be proving the obvious, it is still important and might have significant technical implication for visual prothesis and vision system research in the future.

---

> ### Author Response · Authors · 2022-11-17
> **Response**
>
> We thank the reviewer for their time and for their insightful comments and the encouraging remarks. Below we address the concern raised.
>
> **While the technical and conceptual contribution in ML and representational learning are relatively limited, this paper is acceptable for demonstrating or proving that an adaptive gain control mechanism as a front-end of CNN can improve the prediction of RGC's responses in different lighting condition. Even though it might be proving the obvious, it is still important and might have significant technical implication for visual prothesis and vision system research in the future.**
>
>
> We agree that our paper is not focussed on standard ML benchmark tasks/tests that would signify our model’s contribution to ML and representational learning. This is because our ML model is primarily for applications in neuroscience, specifically for predicting neural activity. The problem we tried addressing in our paper is that current ML models fail to predict neural responses, in this case of retina, to the same spatio-temporal stimulus but under different ambient conditions. This is a challenging problem when building ML models of neurons because the features that the retinal output represents, changes with ambient conditions. As far as we know, there are no existing benchmarks for such a task. In our paper, we present one way of dealing with this problem by equipping ML models with photoreceptor adaptive mechanisms that we know contribute, at least partially, to the change in feature representation at the retinal output. We therefore believe that our work also significantly contributes to ML and representational learning by way of embedding biologically-inspired dynamics in ML models.

---

### Official Review · Reviewer_b3Ty · 2022-10-25

**Confidence:** 3
**Correctness:** 3
**Technical Novelty And Significance:** 3
**Empirical Novelty And Significance:** 3
**Recommendation:** 6

**Clarity, Quality, Novelty And Reproducibility:**

Combining a biophysical photoreceptor frontend with a standard DNN architecture is novel to my knowledge.

The writing is clear and easy to follow.

As noted above, I do not believe the paper reports any statistics or measures of confidence…

The cross-validation approach used to evaluate generalization to extreme lighting conditions is convoluted but seems sound.

The authors do not report any data on whether training the photoreceptor model is important or not. How did the authors choose which parameters to optimize? The photoreceptor model has a small number of parameters compared with the DNN layers, and this mismatch might make it hard to jointly train using first-order optimization algorithms such as ADAM.

It is not clear how the L1 and L2 penalties were selected (grid search?). Were the hyper-parameters chosen using the same test data used to compute explained variance? If so, this would introduce an upward bias in prediction accuracy. It would be preferable to select hyper-parameters using validation and measure performance in a separate test data set.

The authors split the trials into 2 when estimating the noise variance. However, I believe they use all the data when estimating prediction accuracy, which produces a mismatch. This mismatch is unlikely to matter since the authors report that the responses were highly reliable, but it would be better to use the same amount of data when evaluating predictions and noise variance or to correct for the extra data in the predictions.

Am I correct that the data reported is from a single rat and a single monkey?

I do not have the expertise to evaluate whether the retinal model is properly implemented.


**Strength And Weaknesses:**

Strengths

I found the paper clear and easy to follow. The results are simple and straightforward.

The paper provides decent evidence that the photoreceptor model improves generalization to new lighting levels, which is an important form of variation.

Exploiting insights from biology to improve machine learning is rare and potentially exciting contribution.

Weaknesses

Applications to vision and ML are preliminary and left to the appendix (Appendix C), which seems likely to limit the impact of the work to the ICLR community. The photoreceptor model is rather complex which I worry will limit its popularity and impact. The authors toy with creating a simpler, convolutional layer based on their model, but again this part of the paper is preliminary and left to the appendix.

The generalization effect in monkey RGCs is only substantial for the case when training on high light levels and testing on low light levels. No measures of statistical significance or confidence are reported.

It would be interesting to know whether the model would have difficulty generalizing to a variety of lighting conditions when trained using naturalistic variation in retinal stimulation. It would also be useful to quantify model performance using naturalistic stimulation since one of the benefits of having a DNN model is that it can potentially predict neural responses to real-world stimuli, as opposed to just white noise.

If one does use a training set with a wide range of lighting levels is there any benefit of the photoreceptor model? For example, does the model require less training time to reach a given level of accuracy? For ML applications, it’s not clear whether the benefit of the photoreceptor model would be worth the added complexity, particularly if it’s relatively easy to create robustness by applying a bit more diversity to the training regime.


**Summary Of The Paper:**

This paper implements a photoreceptor model in Keras and uses it as a front end in training a shallow CNN model to predict retinal responses to white noise stimuli. The paper reports that the photoreceptor model allows the model to generalize better to new lighting levels. This observation is demonstrated using monkey and rat RGC data. The light levels in the rat experiments used a more dramatic range of variation (4 orders of magnitude) that required a more complex cross-validation procedure due to largely non-overlapping rod and cone activation across levels.

**Summary Of The Review:**

The combination of a photoreceptor frontend with a DNN to predict RGC responses is novel and potentially interesting. I am overall convinced that this addition helps with generalization to novel lighting levels, at least when the differences are extreme. It is unclear to me whether this insight will in practice be useful to the ML community. There might be simple training strategies that would avoid this issue, and the photoreceptor model is complex and seems unlikely to become a mainstay of computer vision models.

Response after rebuttal:

I apologize for my slow response. The authors have addressed all of my comments, which I appreciate. I think this paper provides a solid contribution to the computational neuroscience literature and seems well-suited to the Neuroscience and Cognitive Science section. I still find the generalization results in monkey RGC a little underwhelming, and my best guess is that the impact of the paper will be modest. Nonetheless, I have increased my score from 5 to 6 based on the authors' argument that their paper is appropriate for the noted section.

---

> ### Author Response · Authors · 2022-11-17
> **Responses 1/3**
>
> We thank the reviewer for their time and for their insightful comments on the strengths and weaknesses of the paper. We have updated the manuscript to overcome the weaknesses. Below we provide a point-by-point response to the concerns raised.
>
> **Applications to vision and ML are preliminary and left to the appendix (Appendix C), which seems likely to limit the impact of the work to the ICLR community.**
>
> We agree that the application of our model to computer vision was left to the appendix. This is because our new model is a novel application of ML to neuroscience, enabling us to predict retina responses under dynamic lighting conditions. This is an unsolved real-world challenge in neuroscience because the neural representation of a stimulus constantly changes as a function of ambient conditions. Therefore, we focused the paper on describing this model as a neural predictor rather than computer vision. Thus, we believe that our work addresses an important real-world neuroscience problem and that it is a good fit for the “Neuroscience and Cognitive Science” section of the ICLR. However we also realize the potential of this work beyond neuroscience and therefore included in the appendix a proof-of-concept that our model could be used in computer vision applications.
>
>
> **The photoreceptor model is rather complex which I worry will limit its popularity and impact. The authors toy with creating a simpler, convolutional layer based on their model, but again this part of the paper is preliminary and left to the appendix.**
>
> The biophysical photoreceptor model serves two main purposes. Using this model in a deep neural network framework allows the network to generalize across light conditions not seen during the training. This will be useful for developing applications like visual prosthetics or using our retina-inspired models as a front-end for cortical models. Secondly, the photoreceptor model, while complex, has parameters that map directly onto the biology. This makes the model more interpretable and will help the neurobiology community to use this model to determine the specific cellular mechanisms that enable the retina to function under a wide range of dynamic light conditions. One could instead use the phenomenological model of a photoreceptor that is much simpler and tractable (and which we include as an appendix in our paper), but it will restrict such biological investigations. We have revised the text of our paper to clarify the motivation (increased biological interpretability) for using the more complex photoreceptor model as opposed to the simpler phenomenological model.
>
> Nonetheless, we do understand the motivation for developing simpler models for the wider ML community that might not be interested in biological investigations, but could still benefit from the photoreceptors’ light adaptation mechanisms. Towards this end, we have developed an adaptive convolutional layer based upon a simplified photoreceptor model and demonstrated a proof-of-concept that it could be used in ML applications beyond neuroscience. This preliminary proof-of-concept demonstration is not the main focus of the paper, and it is presented in Appendix.
>
>
> **The generalization effect in monkey RGCs is only substantial for the case when training on high light levels and testing on low light levels. No measures of statistical significance or confidence are reported.**
>
> We have now included the confidence intervals. We would also like to highlight that in revising the paper, we improved the input normalization procedure which makes all models perform slightly better. The normalization procedure is also described in more detail now.
>
>
> **It would be interesting to know whether the model would have difficulty generalizing to a variety of lighting conditions when trained using naturalistic variation in retinal stimulation. It would also be useful to quantify model performance using naturalistic stimulation since one of the benefits of having a DNN model is that it can potentially predict neural responses to real-world stimuli, as opposed to just white noise.**
>
> That is indeed our aim. However, most retina data available either contains experiments using natural stimuli at just a single light level to study retinal responses to natural stimuli or contains experiments using artificial stimuli like white noise at multiple light levels to investigate the effect of ambient light levels on retinal response. To test the model’s generalizability across light levels, we need retina responses to the same stimuli at different light conditions. We are currently working on those experiments, and we plan to use natural stimuli in this future work.

---

> ### Author Response · Authors · 2022-11-17
> **Responses 2/3**
>
> **If one does use a training set with a wide range of lighting levels is there any benefit of the photoreceptor model? For example, does the model require less training time to reach a given level of accuracy? For ML applications, it’s not clear whether the benefit of the photoreceptor model would be worth the added complexity, particularly if it’s relatively easy to create robustness by applying a bit more diversity to the training regime.**
>
> In ML for computer vision the training set can be diversified by varying conditions such as image contrast, orientation, scale, etc. However, in ML for neuroscience, obtaining diverse training sets is more challenging. Input data cannot be augmented prior to training as the labels (neural responses) change across different input conditions. This means that neural data has to be collected at different conditions with several repeats at each condition (to average out noise). This is both experimentally and computationally expensive, which limits the diversity of lighting conditions for which data can be obtained. This issue is compounded by the fact that very long retinal recordings are impractical due to tissue degeneration and other biological/technical factors. Embedding neural mechanisms within the ML framework is one way to overcome the need for training at many different conditions, dramatically reducing the training data requirements. For example, with the photoreceptor mechanisms, the model can generalize to a light level not seen during training. We thus believe that our work is a promising step towards simplifying the data requirements for making neural predictors that generalize across conditions.
>
> Notably, we have also developed a simpler phenomenological layer (the adaptive convolution layer) that can be used in place of the photoreceptor layer. This layer loses the biological interpretability of the photoreceptor layer, but is simpler. It could thus be useful in applications where the biological photoreceptor details are not of interest, and the aim is simply to approximate the computation in the simplest way possible.
>
>
> **As noted above, I do not believe the paper reports any statistics or measures of confidence.**
>
> We have now included measures of confidence.
>
>
> **The authors do not report any data on whether training the photoreceptor model is important or not. How did the authors choose which parameters to optimize? The photoreceptor model has a small number of parameters compared with the DNN layers, and this mismatch might make it hard to jointly train using first-order optimization algorithms such as ADAM.**
>
> As pointed out, the mismatch makes it hard to jointly train the model. Therefore, we set the initial values of all photoreceptor (PR) parameters to their known values from experimental fits. The combined model performs ok at these default values but the performance is improved by training the photoreceptor model using backprop as part of the end-to-end training of the DNN. The trainable parameters in the PR model are the rate and the gain factors of the molecular processes such as the opsin gain and activation, the phosphodiesterase activation and decay, and the rate of calcium removal. These factors strongly govern the gain and temporal kinetics with which photons are converted into electrical activity and are different across rod and cone photoreceptors. Other parameters such as concentration of cGMP in darkness and factors governing cGMP conversion into current, etc, are similar across photoreceptor types and therefore were set to their known values from experimental fits and not trained. We did experiment with making more parameters trainable but it did not improve the model’s performance. We have now included these details in the paper.
>
>
> **It is not clear how the L1 and L2 penalties were selected (grid search?). Were the hyper-parameters chosen using the same test data used to compute explained variance? If so, this would introduce an upward bias in prediction accuracy. It would be preferable to select hyper-parameters using validation and measure performance in a separate test data set.**
>
> We selected L2 weight penalties at each layer to avoid loss of information and be more robust to outliers. L1 penalty was applied to the output of the dense layer because the neural activity itself is relatively sparse and L1 penalties are known to induce sparsity. The test dataset at the new light level – used for testing the generalization performance – was not used in training or in hyperparameter optimization. Early-stopping of the model was based on a validation dataset separate from the test dataset used for measuring the model’s performance. We have updated the text to explain these details.

---

> ### Author Response · Authors · 2022-11-17
> **Responses 3/3**
>
> **The authors split the trials into 2 when estimating the noise variance. However, I believe they use all the data when estimating prediction accuracy, which produces a mismatch. This mismatch is unlikely to matter since the authors report that the responses were highly reliable, but it would be better to use the same amount of data when evaluating predictions and noise variance or to correct for the extra data in the predictions.**
>
> We use the same amount of data when estimating prediction accuracy, i.e. half of the available trials. The trials that go into this half are selected randomly. We have updated the text to explain this in more detail.
>
> **Am I correct that the data reported is from a single rat and a single monkey?**
>
> The paper includes data from 2 rats and 1 monkey.

---

### Author Response · Authors · 2022-11-17
**Response to the main concern raised by all the reviewers**

We thank the reviewers for their insightful comments on the strengths and weaknesses of the paper. We have updated the manuscript to overcome the weaknesses. The main criticism from all the reviewers was that we do not apply our model to real-world problems or to standard ML benchmark tasks such as computer vision. For that reason, the reviewers argue that the paper might not be suited for the ICLR audience.

While we agree that our paper is not focussed on standard ML benchmark tasks, we disagree with the reviewers’ assessment that we have not studied real-world problems, and we disagree with their assessment about the suitability of our work to the ICLR audience. Our new class of ML model is a novel application of ML to neuroscience, enabling us to predict retina responses under dynamic lighting conditions. This is an unsolved real-world challenge in neuroscience because the neural representation of a stimulus constantly changes as a function of ambient conditions. It is also an important practical challenge for the development of sight-restoring retinal prosthetics. Our new ML model is able to learn the retina’s dynamic video representations in these variable conditions whereas the pre-existing CNN models could not. Thus, we believe that our work addresses an important real-world neuroscience problem and that it is a good fit for the “Neuroscience and Cognitive Science” section of the International Conference on Learning Representations.

While we disagreed with the reviewers’ main conclusions, their comments on this point were very helpful because they emphasized to us that we had not clearly enough explained the real-world problem that our work addresses. We will revise the paper to make the main contribution more clear: we developed a novel ML system to solve a challenging real-world problem in the neuroscience of visual representations. With that in mind, we humbly request that the reviewers and the area chair reconsider their assessment of the suitability of our work for ICLR.

Beyond the “Neuroscience and Cognitive Science” section, we anticipate that our work could be of interest to the broader ICLR audience because equipping ML models with adaptive mechanisms like our fully-trainable photoreceptor layer enables those models to account for changes in target labels (in our case, neural responses) that depend on the input statistics (in our case, mean light level). This opens up the possibility of creating artificial neurons capable of adapting easily to complex real-world dynamics. We demonstrated the potential application of this idea to computer vision as a very small-scale proof-of-concept. Because applications to computer vision were not the main purpose of the paper, this demonstration was limited to the appendix section.

Key changes to the paper:
- Edits addressing reviewer comments
- Updated Fig. 3 and Fig. 4 to reflect better data normalization that overall improves the performance of all models. The normalization is explained in the text. Overall, the conclusions remain the same.
- All major text edits are in blue color

---

### Decision · Program_Chairs · 2023-01-20

**Decision:**

Reject

**Justification For Why Not Higher Score:**

I think the core biophysically-inspired layer is a reasonable contribution.  I think there are some legitimate concerns about this contribution in its own right raised by the reviews as well as questions about the impact of this innovation even within computational neuroscience.  On top of this, the authors don't seem to appreciate that their paper actually does make claims (even if they are secondary) about the utility of their innovation for CV settings that multiple reviewers legitimately question in their reviews.

**Justification For Why Not Lower Score:**

N/A

**Metareview: Summary, Strengths And Weaknesses:**

This work Introduces a new photoreceptor-inspired CNN layer that mimics adaptation to light level of biological cells in the retina.  While the direct focus of the work is on improving biological modeling, a secondary motivation is to develop approaches that will improve performance of computer vision algorithms. The novel layer is a biophysically-inspired adaptation model that is implemented as a differentiable layer in Keras.  This method outperforms CNN baselines on settings with variable lighting conditions.

Two out of three reviewers initially leaned negative, but one of the reviewers adjusted their score in light of author responses.  Reviewer b3Ty initially gave a score of 5 and found the paper clear and technically sound in terms of the computational neuroscience contribution.  Author responses prompted this reviewer to raise their score to a 6, but they remained underwhelmed by the results and didn't feel it would be impactful.  Reviewer TU1Q found the core innovation sensible and the empirical validation sufficient, but noted the technical and conceptual contributions are limited from an ML perspective.  Reviewer Rcws initially read the paper primarily from the perspective of a bio-inspired solution to a computer vision problem, rather than as a computational neuroscience effort in its own right.  In response to this, the authors made a reasonable request for the AC to clarify the suitability of their submission.  I replied both to the authors and to the reviewer, indicating my agreement with the authors that the topic matter is suitable for an ICLR paper.

However, in the authors' "Response to the main concern raised by all the reviewers" as well as in a message to the AC, the authors oversimplify the more nuanced concerns of the reviewers, incorrectly asserting that all reviewers are improperly evaluating the paper based on topic. For example, both reviewers Rcws and b3Ty completely reasonably raised a concern about the evaluation only being performed on white noise stimuli.  While the authors replied that white noise data is standard in neuroscience, I don't really find this acceptable given that the authors had considerable flexibility and collected their own data for this paper (i.e. "monkey data were collected by us as part of this study"). Another paper that was itself accepted to ICLR back in 2017 entitled "Multilayer Recurrent Network Models of Primate Retinal Ganglion Cell Responses" also looks at modeling primate retinal cells and involves presentation of natural images to the retina, further indicating the authors' claims that white noise is sufficient isn't entirely consistent with what has been previously studied. That paper focuses on RNNs to capture adaptation phenomena, which also points to the possibility that RNNs of some sort might be an alternative to the biophysically inspired adaptation layer (a baseline not considered in the present work).

In addition, there remains a bit of a confusingly split motivation to the paper as it was written.  Two reviewers pointed out in their reviews some version of confusion as to whether the approach introduced here for modeling biology is a good idea for computer vision.  Reviewer b3Ty noted that applications are only covered in the appendix and wondered why not train models across multiple light levels.  Reviewer Rcws pointed out that for computer vision applications, high-dynamic range data could be a more practical choice for dealing with light variation.  The authors responded in multiple places that this work is focused on predicting neural data as the application.  While, as noted previously, I agree that neural data prediction is a completely valid problem space, I believe the authors invited this confusion by making claims that their method may be useful for computer vision applications.  For example, the abstract states, "Such processing capabilities are desirable in many settings, including computer vision systems that operate in dynamic lighting environments like in self-driving cars, and in algorithms that translate visual inputs into neural signals for use in vision-restoring prosthetics."  It isn't clear to me that self-driving cars and retinal prosthetics are really facing the same challenge.  And the discussion + appendix further muddle the message by making claims about the CV-ulitity of the approach with only weak support.  Indeed the author "Response to the main concern raised by all the reviewers" doubles down on the claims of utility for CV applications while at the same time claiming the paper is being unfairly evaluated.